# Light Therapy for Cancer-Related Fatigue in (Non-)Hodgkin Lymphoma Survivors: Results of a Randomized Controlled Trial

**DOI:** 10.3390/cancers13194948

**Published:** 2021-09-30

**Authors:** Daniëlle E. J. Starreveld, Laurien A. Daniels, Jacobien M. Kieffer, Heiddis B. Valdimarsdottir, Jessie de Geus, Mirthe Lanfermeijer, Eus J. W. van Someren, G. Esther A. Habers, Jos A. Bosch, Cécile P. M. Janus, Dick Johan van Spronsen, Roel J. de Weijer, Erik W. A. Marijt, Eva de Jongh, Josée M. Zijlstra, Lara H. Böhmer, Margreet Houmes, Marie José Kersten, Catharina M. Korse, Huub H. van Rossum, William H. Redd, Susan K Lutgendorf, Sonia Ancoli-Israel, Flora E. van Leeuwen, Eveline M. A. Bleiker

**Affiliations:** 1Department of Psychosocial Research and Epidemiology, The Netherlands Cancer Institute, 1066 CX Amsterdam, The Netherlands; d.starreveld@nki.nl (D.E.J.S.); j.kieffer@nki.nl (J.M.K.); jessiedegeus@gmail.com (J.d.G.); f.v.leeuwen@nki.nl (F.E.v.L.); 2Department of Radiotherapy, Amsterdam University Medical Center, 1105 AZ Amsterdam, The Netherlands; l.a.daniels@amsterdamumc.nl; 3Department of Oncological Sciences, Mount Sinai School of Medicine, New York, NY 10029, USA; heiddisb@ru.is (H.B.V.); william.redd@mssm.edu (W.H.R.); 4Department of Laboratory Medicine, The Netherlands Cancer Institute, 1066 CX Amsterdam, The Netherlands; m.lanfermeijer@nki.nl (M.L.); t.korse@nki.nl (C.M.K.); h.v.rossum@nki.nl (H.H.v.R.); 5Department of Sleep and Cognition, Netherlands Institute for Neuroscience (An Institute of the Royal Netherlands Academy of Arts and Sciences), 1105 BA Amsterdam, The Netherlands; e.van.someren@nin.knaw.nl; 6Department of Psychiatry, Amsterdam University Medical Center, 1007 MB Amsterdam, The Netherlands; 7Faculty of Science, Integrative Neurophysiology, Vrije Universiteit Amsterdam, 1081 HV Amsterdam, The Netherlands; 8Health, Medical, and Neuropsychology Unit, Institute of Psychology, Leiden University, 2333 AK Leiden, The Netherlands; g.e.a.habers@fsw.leidenuniv.nl; 9Department of Clinical Psychology, University of Amsterdam, 1018 WV Amsterdam, The Netherlands; j.a.bosch@uva.nl; 10Department of Radiation Oncology, Erasmus MC Cancer Institute, 3015 GD Rotterdam, The Netherlands; c.janus@erasmusmc.nl; 11Department of Hematology, Radboud University Medical Center, 6525 GA Nijmegen, The Netherlands; Dick-Johan.vanSpronsen@radboudumc.nl; 12Department of Hematology, University Medical Center Utrecht Cancer Center, 3508 GA Utrecht, The Netherlands; R.deWeijer@umcutrecht.nl; 13Department of Hematology, Leiden University Medical Center, 2333 ZA Leiden, The Netherlands; W.A.F.Marijt@lumc.nl; 14Department of Hematology, Albert Schweitzer Hospital, 3318 AT Dordrecht, The Netherlands; E.deJongh@asz.nl; 15Department of Hematology, Amsterdam University Medical Centers, Cancer Center Amsterdam, 1081 HV Amsterdam, The Netherlands; J.Zijlstra@amsterdamumc.nl; 16Department of Hematology, Haga Teaching Hospital, 2545 AA The Hague, The Netherlands; l.h.bohmer@hagaziekenhuis.nl; 17Department of Hematology, Admiraal de Ruyter Hospital, 4462 RA Goes, The Netherlands; margreet.houmes@adrz.nl; 18Department of Hematology, Amsterdam University Medical Centers, Cancer Center Amsterdam, 1105 AZ Amsterdam, The Netherlands; m.j.kersten@amsterdamumc.nl; 19Departments of Psychological and Brain Sciences, Obstetrics and Gynecology, Urology, Holden Comprehensive Cancer Center, University of Iowa, Iowa City, IA 52240, USA; susan-lutgendorf@uiowa.edu; 20Department of Psychiatry, University of California, San Diego, CA 92093-0737, USA; sancoliisrael@health.ucsd.edu; 21Department of Clinical Genetics, Leiden University Medical Center, 2333 ZA Leiden, The Netherlands

**Keywords:** cancer-related fatigue, light therapy, circadian rhythms, sleep, randomized controlled trial

## Abstract

**Simple Summary:**

Cancer-related fatigue (CRF) is one of the most frequently reported symptoms with prevalence rates of 25 to 60 percent in (non-)Hodgkin lymphoma survivors. Several (pilot) studies showed promising effects of light therapy to reduce CRF. The aim of the current study is to evaluate the short- and long-term efficacy of light therapy on CRF and associated symptoms in chronically fatigued (non-)Hodgkin lymphoma survivors. Eighty-three survivors were exposed to bright white light (intervention) and another 83 survivors were exposed to dim white light (comparison). Results showed that all participants, irrespective of light condition, reported reduced levels of fatigue after the completion of light therapy. Similar results were found for depression, sleep quality, and some aspects of quality of life. No effect was found on circadian rhythms or objectively assessed sleep. Therefore, it is important to further investigate which aspects of intervention are associated with the improvements observed after light therapy.

**Abstract:**

Purpose: To evaluate the short- and long-term effects of light therapy on fatigue (primary outcome) and sleep quality, depression, anxiety, quality of life, and circadian rhythms (secondary outcomes) in survivors of (non-)Hodgkin lymphoma presenting with chronic cancer-related fatigue. Methods: We randomly assigned 166 survivors (mean survival 13 years) to a bright white light intervention (BWL) or dim white light comparison (DWL) group. Measurements were completed at baseline (T0), post-intervention (T1), at three (T2), and nine (T3) months follow-up. A mixed-effect modeling approach was used to compare linear and non-linear effects of time between groups. Results: There were no significant differences between BWL and DWL in the reduction in fatigue over time. Both BWL and DWL significantly (*p* < 0.001) improved fatigue levels during the intervention followed by a slight reduction in this effect during follow-up (ES_T0-T1_ = −0.71; ES_T1-T3_ = 0.15). Similar results were found for depression, sleep quality, and some aspects of quality of life. Light therapy had no effect on circadian rhythms. Conclusions: BWL was not superior in reducing fatigue compared to DWL in HL and DLBCL survivors. Remarkably, the total sample showed clinically relevant and persistent improvements on fatigue not commonly seen in longitudinal observational studies in these survivors.

## 1. Introduction

Cancer-related fatigue (CRF) is one of the most frequently reported symptoms with prevalence rates of 25 to 60 percent in survivors of Hodgkin lymphoma (HL) and diffuse large B-cell lymphoma (DLBCL) [1,2,3,4]. CRF is related to a lower quality of life and often described as part of a symptom cluster, including sleep disturbances, depression, anxiety, and pain [1,5,6,7,8,9]. In cancer patients, these symptoms are associated with circadian disruptions, e.g., more sleep disruptions during the night and/or napping during the day [10,11,12,13,14,15]. Light therapy, in which individuals are exposed to bright light, is known for its positive effect on seasonal affective disorders and circadian rhythm disorders [16,17,18,19,20]. It is assumed to work via its restorative effect on circadian rhythms through stimulation of the suprachiasmatic nucleus (the biological clock), although other mechanisms of action, for example, the stimulation of mood regulation areas, have also been reported [21,22,23].

Three studies showed promising results of morning bright light therapy as a treatment for CRF in cancer patients undergoing chemotherapy and in cancer survivors [24,25,26]. These results also suggested that light therapy improved sleep quality, quality of life, and restored circadian sleep–wake cycles [26,27,28,29,30,31]. However, these studies had several methodological limitations, including small sample sizes and short follow-up assessments (3 weeks post-intervention) [24,25,26].

Therefore, the present study investigated the effect of light therapy on CRF in a randomized controlled trial in a large sample of cancer survivors with a follow-up of 9 months. The primary aim was to investigate the short- and long-term efficacy of light therapy in decreasing CRF and improving sleep quality, depression, anxiety, quality of life, and circadian disruptions in HL and DLBCL survivors with CRF. We hypothesized that participants exposed to bright white light (BWL), the intervention group, would show an improvement in fatigue compared to participants exposed to dim white light (DWL), the comparison group. Secondly, we expected improvements in associated symptoms, including sleep quality, depression, anxiety, quality of life, and entrainment of circadian rhythms.

## 2. Patients and Methods

### 2.1. Research Design and Study Sample

The study design of this double-blind, randomized controlled trial has been described in detail elsewhere [32]. Briefly, survivors with a history of lymphoma were recruited from ten hospitals in the Netherlands. Inclusion criteria were: (1) age between 18 and 70 years; (2) primary diagnosis of HL or DLBCL at least 2 years prior to study entry; (3) moderate to severe fatigue since diagnosis and/or treatment. Exclusion criteria covered other factors that could have affected acute fatigue or circadian rhythms. The study was approved by the institutional review board of the Netherlands Cancer Institute (number NL61017.031.17) and all participating hospitals, and is registered at ClinicalTrials.gov (NCT03242902).

### 2.2. Procedure, Randomization, and Timing of Assessments

Participant enrollment took place between September 2017 and October 2019. Figure 1 provides the CONSORT diagram. Briefly, survivors were recruited via referrals from clinicians or through participation in a survey study on bedtime, sleep quality, and CRF [33]. Survivors received an information brochure, screening questionnaire, and response card to indicate interest in participation, or reasons for non-participation. Interested survivors were screened by telephone to confirm eligibility. Eligible survivors received a patient information letter.

After providing written informed consent, participants were randomly assigned to the BWL or DWL group at a 1:1 ratio, stratified by diagnosis, time since diagnosis, and gender, by a research assistant not involved in the study. All other study personnel were blinded to the condition until a participant had completed the final assessment. Participants were informed that two intensities of light therapy were being compared without being informed regarding the hypotheses.

Participants were assessed at baseline (T0), after 25 days of light therapy (T1), and at three (T2), and nine months (T3) after treatment. T0 and T1 included a visit to the hospital to provide instructions and exchange study materials. T2 and T3 were completed at home. After completion of T3, participants received information on their assigned condition.

### 2.3. Intervention

In line with previous studies, the first 37 participants used the Litebook Edge (Litebook, Ltc. Medicine Hat, AB, Canada) [24,25]. Confirmatory spectral measurements of the Litebook established a light intensity of 351 lux at eye level for the BWL condition. As this is comparable to ‘office lighting’ and may not be sufficient for light therapy, we changed to Luminette glasses (Lucimed SA, Villers-le-Bouillet, Belgium). This light source exposed individuals to broad-spectrum white light, enriched at 468 and 570 nm of 1.500 lux at eye level for BWL, and 8 lux for DWL (see Appendix A). All participants, including the Litebook users, were included in the intention-to-treat analyses.

The light therapy protocol, based on previous studies, instructed participants to use light therapy for 30 min, daily, within 30 min after awakening, for a duration of 25 days at home [24,25]. Other activities, such as reading or having breakfast, were permitted during therapy. A member of the research staff called on the fifth day to check for side effects.

### 2.4. Study Measures

Sociodemographic information was collected with the screening and baseline questionnaire. Clinical information was abstracted from the patient’s medical records. Primary outcomes included general fatigue, assessed by the visual analogue scale (VAS)-fatigue [34], from 0 (no fatigue) to 10 (worst imaginable fatigue), Multidimensional Fatigue Inventory (MFI), general fatigue scale [35,36]), and restrictions caused by fatigue (Works and Social Adjustment Scale (WSAS) [37]).

Secondary outcomes included questionnaires to assess sleep quality (Pittsburg Sleep Quality Index (PSQI) [38]), depression (Center for Epidemiological Studies—depression scale (CES-D) [39]), anxiety (State Trait Anxiety Inventory—6 items (STAI-6) [40]), quality of life (RAND 36-item Health Survey (RAND-36) [41,42]), assessments of sleep (wrist actigraphy [43,44]), salivary concentrations of cortisol [45], and melatonin [46,47] (see Appendix A). A detailed description of the outcomes is provided in Table 1.

### 2.5. Statistical Analyses

With 64 participants per group, the study had an 80% power to detect an effect size (ES) of 0.50 for the main effect of light therapy on fatigue with a two-tailed *p*-value of 0.05. Cohen’s effect size of 0.5 means a 0.5 standard deviation difference on the primary measurement outcome, which is considered to be a meaningful clinical difference [48]. Thirty-seven additional participants were recruited to ensure sufficient power for the sensitivity analyses in Luminette users. Comparisons of baseline characteristics between groups were performed using independent samples *t*-test (continuous variables) or chi-square (categorical variables), or, in case of unmet assumptions, Mann–Whitney (continuous variables), or Fisher’s Exact (categorical variables) tests. Scores on patient-reported outcome measures were calculated according to published algorithms. In case of missing values on single items in a questionnaire, the missing values were replaced by the average score of the completed items in the same scale for each individual, provided that at least 50% of the items of a scale had been completed. For example, if one item of the PSQI was not completed, we calculated the average score on that specific scale at that specific measurement point of that participant and replaced the missing value with this score. If more than 50% of the items of a scale were missing, the score on that scale was considered missing.

To evaluate differences between the groups over time in primary and secondary outcomes, we used a mixed effect modeling approach with random intercept and slope with a maximum likelihood solution. We modeled linear and quadratic time effects to determine if an initial change in the outcome was maintained during follow-up. The choice for models with linear or non-linear effects, for models with different covariance structures (UN, AR1, CS), and models corrected for potential non-ignorable dropouts were determined by using the Bayesian Information Criterion (BIC) and the Akaike’s Information Criterion (AIC) [49,50]. The overall mean change and difference in mean change scores over time between groups during the active treatment phase (T0-T1) and follow-up period (T1-T3) were accompanied by standardized effect sizes (ES) calculated based on the estimated marginal means and pooled SD: (mean_T1_-mean_T0_)/pooled SD_T0-T1_ or (mean_T3_-mean_T1_)/pooled SD_T1-T3_. ESs of 0.20 were considered small, 0.50 moderate, and 0.80 large [51]. To limit type-I errors due to multiple testing, a *p*-value of 0.01 was considered statistically significant.

At the individual patient level, clinically relevant improvement was determined on a 1.1-point decrease on the VAS-fatigue, a 2.0-point decrease on the general fatigue subscale of the MFI, or a 4.1-point decrease (0.5 standard deviation) on the WSAS [48,52,53,54,55]. ꭕ^2^ tests were used to compare differences in improvement between the intervention and comparison group.

All analyses were conducted on an intention-to-treat (ITT) basis. Additionally, we performed one per-protocol analysis including participants who used light therapy on all 25 treatment days and two sensitivity analyses on data from participants who used (1) Luminette glasses; and (2) light therapy during autumn/winter (October to March). All statistical analyses were conducted in SPSS version 25.

## 3. Results

In total, 984 survivors were invited to participate in the study, of whom 321 (33%) returned a response card indicating that they were not interested, and 309 (31%) did not respond (Figure 1).

Of the 354 interested survivors, 273 (77%) survivors met criteria for further screening and 211 (60%) were eligible for participation, of whom 170 (48%) signed informed consent. Four participants withdrew informed consent prior to randomization. The remaining 166 participants were randomized to the BWL (*n* = 83) or the DWL (*n* = 83) group and were included in the intention-to-treat analysis. Single items missing values were detected for the PSQI, STAI-6, CWS, MOS-Cog, and MDASI in less than 5% of the participants. Completion rates of self-reported questionnaires at baseline assessment T0 (99%), follow-up assessment T1 (95%), T2 (86%), and T3 (87%) differed significantly between groups at T1 (BWL: 99% *v* DWL: 90%; *p* = 0.03). Correction for non-ignorable dropouts did not improve model fit (Appendix A). The completion of T3 during COVID-19 restrictions (*n* = 33; 23%) did not differ between groups and did not affect the study results (Appendix A). Presented results are uncorrected for these factors. Availability rates of actigraphy-derived sleep and circadian variables at T0 (95% and 94%, respectively), T1 (89% and 87%, respectively), T2 (83% and 81%, respectively), and T3 (84% and 83%, respectively) did not differ between groups. Availability rates of cortisol and melatonin concentrations at T0 (100% and 100%, respectively) and T1 (96% and 93%, respectively) were similar between the groups (see Appendix A).

Most participants were HL survivors (83%). Their mean age was 45.7 years and the average time since lymphoma was 12.9 years. Almost all participants had received chemotherapy (93%) and/or radiotherapy (72%). Baseline levels of fatigue were high (mean VAS-fatigue = 6.1; mean MFI general fatigue = 15.7; mean WSAS = 20.5). Except for marital status (*p* = 0.03), all baseline characteristics were balanced between the groups (see Table 2).

Table 3 shows the characteristics of light therapy use by the participants. According to the light therapy diaries (*n* = 155), 37% used light therapy all 25 days and 56% used light therapy for 14 to 25 days, with a median time between sleep offset and light therapy start of 19 min (range: 5–109 min). In the complete sample (*N* = 166), 13 survivors stopped prematurely with the study. Reasons for attrition were self-reported side effects (*n* = 7), time constraints, or personal circumstances (*n* = 6).

### 3.1. Primary Outcomes 

There were no significant differences between BWL and DWL in the improvement of fatigue over time (Figure 2 and Appendix A). Both BWL and DWL (Appendix A) led to a statistically significant, clinically relevant, improvement of fatigue during the intervention, followed by a slight reduction in this effect during follow-up (VAS fatigue: ES_T0-T1_= −0.71, ES_T1-T3_= 0.15, *p* < 0.001; MFI general fatigue: ES_T0-T1_= −0.81, ES_T1-T3_= 0.13, *p* < 0.001). The improvement of restrictions caused by fatigue showed a moderate effect during the intervention, which further improved slightly during follow-up (WSAS: ES_T0-T1_= −0.32, ES_T1-T3_= −0.07, *p* < 0.001). At an individual level, results showed no differences in the number of participants with clinically relevant improvements on primary outcomes between both groups (Table 4).

### 3.2. Secondary Outcomes 

There were no significant differences between BWL and DWL on secondary outcomes (Figure 2, Appendix A). Both BWL and DWL (Appendix A) led to statistically significant improvements, indicating moderate effects during the intervention followed by a slight reduction in this effect during follow-up, for sleep quality (ES_T0-T1_= −0.44, ES_T1-T3_= 0.10, *p* < 0.001), and depression (ES_T0-T1_= −0.41, ES_T1-T3_= 0.16, *p* = 0.004). Three aspects of health-related quality of life showed statistically significant improvements of moderate effects during the intervention, followed by small further improvements during follow-up: role limitations due to physical functioning (ES_T0-T1_= 0.33, ES_T1-T3_= 0.11, *p* < 0.001), energy (ES_T0-T1_= 0.48, ES_T1-T3_= 0.05, *p* < 0.001), and social functioning (ES_T0-T1_= 0.35, ES_T1-T3_= 0.09, *p* = 0.002). No significant group differences or overall time effects were observed for anxiety, the remaining subscales of the RAND-36, and actigraphy-derived sleep. No effects were observed for cortisol and melatonin (Figure 3, Appendix A).

The per-protocol analysis, including individuals who adhered to 25 days of light therapy, showed similar results except for a group difference in the effect of light therapy on sleep efficiency (Appendix A). Sleep efficiency improved in the BWL group and deteriorated in the DWL group between T2 and T3, suggesting that this effect did not result from light therapy. The sensitivity analyses for individuals who used Luminette glasses or light therapy during autumn/winter yielded similar results (Appendix A).

### 3.3. Adverse Effects 

Two participants were hospitalized for at least one night because of serious adverse events not related to the study (stress-related symptoms and pancreatitis). Self-reported side effects, e.g., headache and/or nausea (22%) and tired eyes (19%), were balanced between groups (Table 2). These effects were temporary and disappeared within five days despite continuation of light therapy.

## 4. Discussion

In this double blind, randomized controlled trial, exposure to morning BWL showed no superiority to morning DWL on fatigue and related symptoms in long-term HL and DLBCL survivors presenting with chronic cancer-related fatigue. Remarkably, both groups showed clinically relevant improvements on fatigue and restrictions caused by fatigue, and improvements on sleep quality, depression, and three aspects of quality of life (role limitations due to physical functioning, energy, and social functioning). This improvement only slightly reduced during follow-up but was still clinically relevant nine months post-intervention. Neither BWL nor DWL had an effect on anxiety, other aspects of quality of life, actigraphy-derived sleep, or cortisol and melatonin concentrations.

In contrast to two earlier studies that investigated the effect of light therapy on cancer-related fatigue in adult cancer survivors, the current larger phase-III trial did not observe superiority of BWL over DWL [25,26]. There were several differences between these studies. Firstly, the average time since diagnosis was much longer in our study (13 years) compared to previous studies (17 months and 28 months). Secondly, previous studies used dim red light (DRL; 50 lux or 400 lux) as a comparison condition, instead of the DWL (20 lux) used in the current trial. An advantage of DWL is that it might be less clear to the participant that he or she is randomized to the comparison condition. However, as the circadian system is most strongly affected by white light enriched around 470 nm, the DWL condition in our study might still have been somewhat effective [22]. This effect is not expected for DRL. Nonetheless, several studies showed that polychromatic light, as used in the current trial, needed an intensity of 393 lux or higher to induce an effect on circadian rhythms [56,57]. This is supported by the study of Valdimarsdottir et al. that showed significant differences in individuals exposed to BWL (1.300 lux) and DWL (90 lux) [58]. It should be noted that the previous study on light therapy for cancer-related fatigue by Johnson et al. only showed superiority of BWL to DRL on the total score of fatigue, with effect sizes of 1.20 and 0.93, respectively, indicating that both groups improved [26]. No superiority of BWL to DRL was reported for five dimensions of fatigue (including general fatigue), mood, depression, quality of life, and sleep quality of which both groups showed improvements, suggesting that the selection of DWL or DRL as comparison may not fully explain the discrepancies between both studies.

It is notable that study participation led to clinically relevant improvements (ES = −0.71 [VAS-fatigue]; ES = −0.81 [MFI general fatigue]) in long-term cancer survivors suffering from chronic fatigue. Although we cannot explain this by differences in light intensity, it is important to further investigate which aspects of the study protocol caused this effect. Firstly, the positive effects might result from lifestyle changes. For example, some participants spontaneously self-reported that they exercised more (36%), which may have increased their light exposure if it was outside, or kept a more regular sleep–wake cycle following light therapy (17%). These activities have been associated with reduced CRF [59,60,61]. Secondly, the improvement might be explained by the personal attention during participation or as a placebo response, which has been reported previously for CRF [62,63]. Thirdly, the decrease in fatigue might reflect a natural improvement over time, although we believe this is unlikely in our study because longitudinal observational studies in long-term cancer survivors showed persisting fatigue [64,65]. Another reason that a natural improvement in the current sample is unexpected is that the participants in the current trial were selected for the presence of long-lasting fatigue since the diagnosis or treatment for cancer. Finally, we cannot exclude the possibility that regression towards the mean explained a small part of the positive effects observed in this trial.

Contrary to our expectations, we found no effect of light therapy on actigraphy-derived sleep or cortisol and melatonin, which follow a circadian rhythm. This is in line with a previous study showing that changes in cortisol levels did not mediate the positive effect of light therapy on CRF in cancer survivors [66]. Moreover, baseline values of actigraphy-derived sleep in the current sample suggest the presence of sleep problems but no circadian disruptions compared to the general population [67,68,69]. Two recent studies also suggested an absence of an association between circadian disruptions and CRF in long-term cancer survivors [33,70]. Therefore, it is unclear whether circadian disruptions are associated with CRF in cancer survivors although research in this group is limited and further exploration is necessary.

Our trial had several limitations. First, two of the three primary outcome measurements (the VAS-fatigue and the MFI) had several limitations. The single item VAS-fatigue measures fatigue as a unidimensional construct. The advantage of this measurement is that it is often used in clinical practice, which improves the interpretation of our results for clinical practice. However, the VAS-fatigue can be influenced by context and other momentary and daily factors [71]. Therefore, we also included the MFI as a primary outcome. This scale had the advantage that the effect of light therapy on different dimensions of fatigue could be determined. While data collection of this trial was ongoing, we performed a psychometric evaluation of the MFI in the general Dutch population [36]. Results showed that the factor structure of the MFI is questionable. The general fatigue subscale was found to be the most reliable part of the MFI and those results are presented in this manuscript. Despite the questionable factor structure, we performed analyses for the total score of the MFI and the remaining original dimensions of fatigue as assessed with the MFI (unpublished work). The results showed an improvement over time for all participants, irrespective of light intensity, for the total score of the MFI, physical fatigue, reduced activity, and reduced motivation. No effect was seen for mental fatigue. This supported the findings observed in the current manuscript.

Secondly, we changed light therapy devices while data collection for this trial was ongoing. This decision was based on the advice from experts in the field to publish a spectral assessment of the light used for light therapy (see Appendix A). Surprisingly, contrary to the information provided by the manufacturer, these measurements showed that the light intensity of the BWL Litebook Edge was not sufficient for the use of light therapy. Therefore, we decided to change to Luminette glasses, which provide light therapy that is more in line with the guidelines of light therapy use in seasonal affective disorders [72,73].

Thirdly, saliva collection for the assessment of circadian rhythms of melatonin and cortisol was performed one day prior to light therapy and one day post-intervention. However, there is day-to-day variance in the assessment of cortisol. For a stable measurement of an individual’s cortisol level and the diurnal slope, it is recommended to collect saliva for at least three consecutive days [74]. Fourthly, we did not include an objective assessment of total daily light exposure. Therefore, we could not confirm self-reported compliance, assess the duration of light therapy, or correct for exposure to natural light. Fifthly, although the compliance rate of 91% in the current study was high compared to previous studies (91% vs. 67–95%, respectively), only a minority (37%) of the participants used light therapy on all 25 days [26,27]. However, the majority (56%) used light therapy for 14–25 days, which is enough to show improvements according to the guidelines of light therapy for SAD [72]. Sixthly, our study sample was limited to (non-)Hodgkin lymphoma survivors, which might reduce generalizability to other populations. However, similar findings are expected in other populations because no associations are reported between fatigue and disease-related factors [75,76,77]. Finally, the number of missing completed questionnaires of the post-intervention measurement differed between groups, with more missing completed questionnaires in the comparison group. However, correction for missing data patterns yielded similar results.

Our study had several strengths, including its multicenter RCT design, larger sample size, high follow-up rates, and the assessment of self-reported as well as behavioral and biological effects of light therapy.

## 5. Conclusions

In conclusion, our data showed no superiority of exposure to BWL compared to DWL. Light therapy, irrespective of light intensity, led to clinically relevant and relatively stable improvements of fatigue, sleep quality, depression, and quality of life in long-term HL and DLBCL survivors with chronic CRF. Therefore, it is important to further investigate which component(s) of the light therapy study protocol explain clinical improvements observed after intervention as well as comparison light conditions.

## Figures and Tables

**Figure 1 cancers-13-04948-f001:**
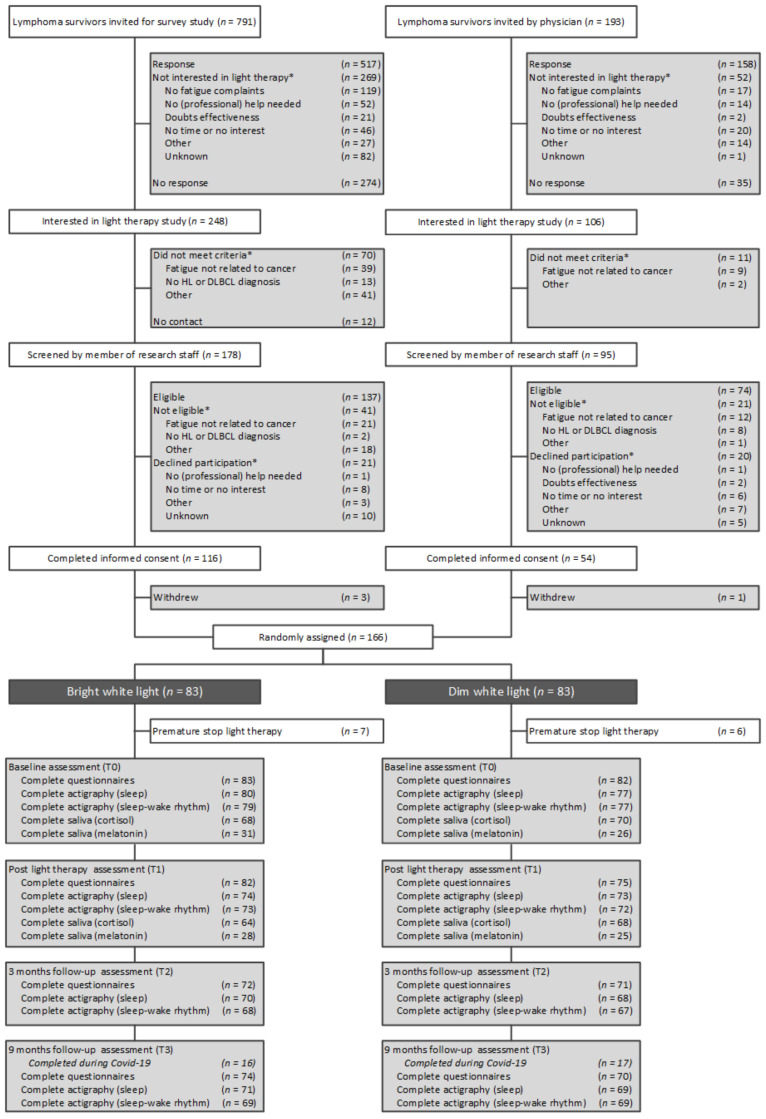
CONSORT diagram. Abbreviations: HL, Hodgkin lymphoma; DLBCL, diffuse large B-cell lymphoma. Note: Number of missing assessments at T1, T2, and T3 were not necessarily cumulative. * Patients could provide more than one reason for non-participation or could be excluded for more than one reason.

**Figure 2 cancers-13-04948-f002:**
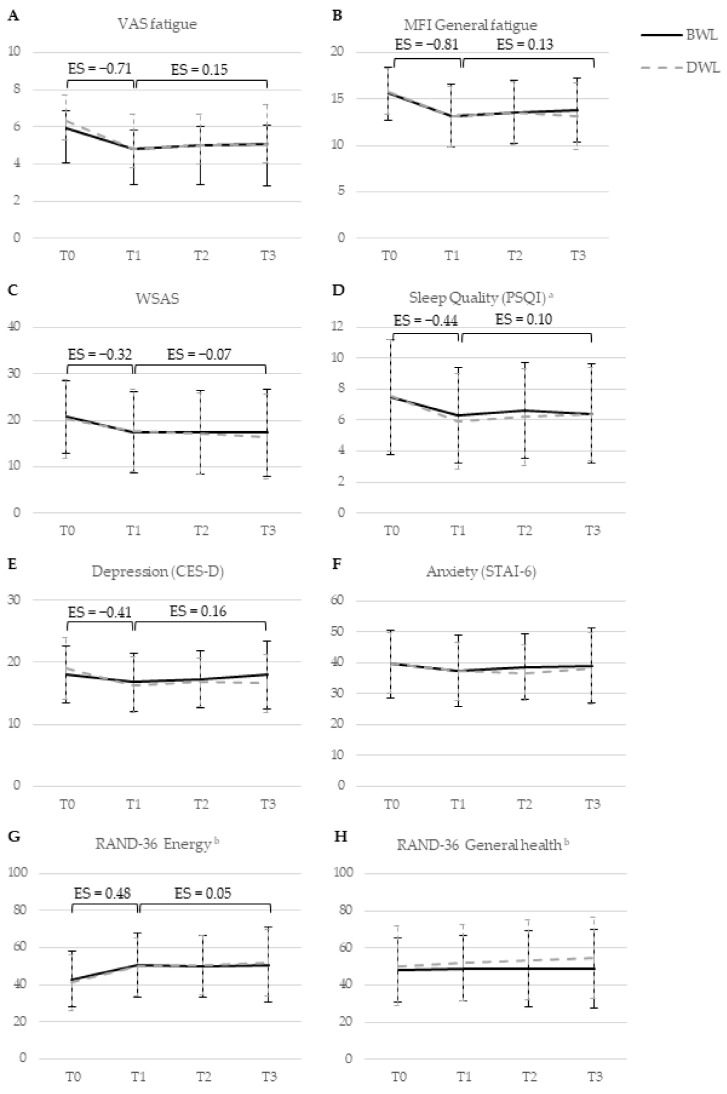
Changes in raw mean levels of primary and secondary self-reported outcomes from baseline to nine months follow-up in groups receiving bright white light therapy (BWL; *n* = 83) and dim white light therapy (DWL; *n* = 83). Bars indicate standard deviations. Effect sizes are shown for significant changes over time in all participants, irrespective of light intensity. Abbreviations: BWL, bright white light; CES-D, Center for Epidemiological Studies—depression; DWL, dim white light; MFI, Multidimensional Fatigue Inventory; PSQI, Pittsburgh Sleep Quality Index; RAND-36, RAND 36-item Health Survey; STAI-6, State Trait Anxiety Index—short form; VAS, fatigue visual analogue scale; WSAS, Work and Social Adjustment Scale. T0, baseline; T1, directly post-intervention; T2, 3 months after the end of light therapy; T3, 9 months after finishing light therapy. ^a^ The total score of the PSQI is shown. Based on the AIC and BIC criteria, the model with the best fit excluded a random slope and included an autoregressive covariance structure. The effect of light therapy on the seven subscales of the PSQI is described in Supplementary material 6. ^b^ The energy and general health subscales of the RAND-36 are shown. The remaining subscales are described in Appendix A.

**Figure 3 cancers-13-04948-f003:**
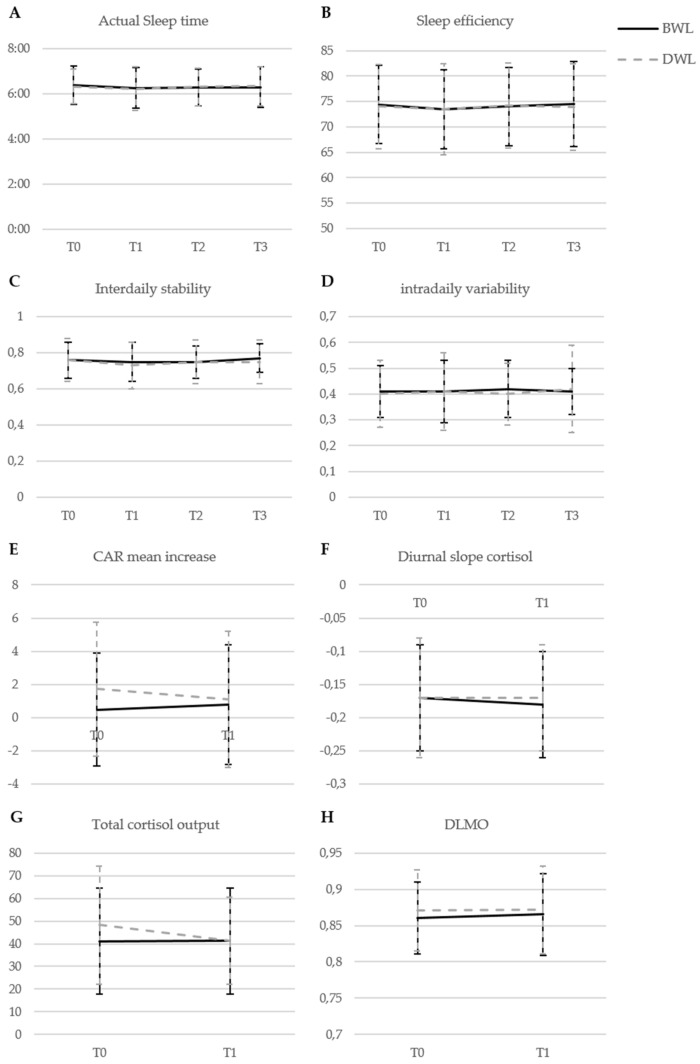
Changes in raw mean levels of actigraphy-derived sleep variables (**A**–**D**), cortisol, and melatonin variables (**E**–**H**) in groups receiving bright white light therapy (BWL; *n* = 83) and dim white light therapy (DWL; *n* = 83). Bars indicate standard deviations. Abbreviations: BWL, bright white light; CAR, cortisol awakening response; DLMO, dim light melatonin onset; DWL, dim white light; T0, baseline; T1, directly post-intervention; T2, 3 months after the end of light therapy; T3, 9 months after finishing light therapy.

**Table 1 cancers-13-04948-t001:** Study outcome measures and corresponding questionnaires.

Variable	Assessment	Details
**Primary Outcomes**		
Cancer-related fatigue	VAS-fatigue	1 item; 11-point Likert scale Total score: 0–10; higher scores indicate more fatigue. Time frame: this moment.
	MFI	20 items; 5-point Likert scale. Subscales: general fatigue, mental fatigue, physical fatigue, reduced motivation, reduced activity. Only general fatigue is used since psychometric validation of this scale indicated that this subscale is the most reliable [36]. Subscale score: 4–20; higher scores indicate more fatigue. Time frame: past few days.
Restrictions caused by fatigue	WSAS	5 items; value range between 0.00 and 8.00. Total score: 0–40; higher scores indicate higher levels of disability. Time frame: influence of fatigue on daily life.
**Secondary Outcomes**		
Sleep quality	PSQI	19 items; 4-point Likert scale and open-ended questions. Subscales: subjective sleep quality, sleep latency, sleep duration, habitual sleep efficiency, sleep disturbances, use of sleeping medication, daytime dysfunction. Total score: 0–21; subscale scores: 0–3; higher scores indicate more acute sleep disturbances. Time frame: past month.
Depression	CES-D	20 items; 4-point Likert scale. Total score: 0–60; higher scores indicate greater depressive symptoms. Time frame: past week.
Anxiety	STAI-6	6 items; 4-point Likert scale. Total score: 20–80; higher scores indicate increased anxiety. Time frame: this moment.
Quality of life	RAND-36	36 items; dichotomous and 3- to 6-point Likert scale. Scales: physical functioning, role limitations due to physical health, role limitations due to emotional problems, energy, emotional well-being, social functioning, pain, general health Scale scores: 0–100; higher scores indicates higher levels of functioning/well-being. Time frame: past 4 weeks.
Sleep	Wrist actigraphy	Device: MotionWatch8 (Camntech, Cambridgeshire, United Kingdom). Software to handle data: MotionWare (Camntech, Cambridgeshire, United Kingdom). Technical settings: epoch length 60 s, tri-axial mode. Location: non-dominant wrist. Time period: 10 days (Friday 18:00 h till Monday 12:00 h). Actigraphy log included: bedtime, attempted time to fall asleep, wake-up time, out-of-bed time, nap times, non-wear times. Derived sleep variables: sleep efficiency, mid sleep, and total bedtime. Derived sleep-wake rhythm variables: Interdaily stability (IS; an estimate of the 24-h sleep-wake rhythm) and intradaily variability (IV; an estimate of the stability of the sleep–wake rhythm) [44]. A measurements point was excluded from the sleep variables analyses when the actigraphy was worn for less than 4 nights and from the sleep–wake rhythm variables analyses when the actigraphy was worn for less than 72 consecutive hours. Scores: IS: 0–2; higher scores indicate a more fragmented rhythm; IV: 0–1; 1 indicates perfect synchronization.
Cortisol	Salivary cortisol	Saliva collection via a passive drool technique in a propylene vial at the participants’ home. Sample collection on five different time points during 24 consecutive hours: (1) at personal waking time, (2) 30 min after awakening, (3) 45 min after awakening, (4) at 16.00 o’clock, and (5) at bedtime. Saliva collection was on the Friday prior to light therapy (start day Monday) and the Friday after completion of light therapy (finish day was Thursday). After sample collection, saliva samples were stored in the refrigerator and mailed to the lab via post where the samples were stored in a freezer at a -80 °C until processing. Cortisol values (nmol/L) were determined using liquid chromatography tandem mass spectrometry. Method imprecisions were ≤13.9% and lower limits of quantitation were 0.5 nmol/L. Derived variables: cortisol awakening response, diurnal cortisol slope, area under the curve. For further details on the analytical method and performance characteristic, see Appendix A.
Melatonin	Salivary melatonin	Subsample (*n* = 60). Collection of five additional saliva samples starting 5 h prior to bedtime followed by one sample every sequential hour. Collection and handling of samples was similar to the procedure described for cortisol. Method imprecisions were ≤11.9% and lower limits of quantitation were 0.01 nmol/L. Derived variables: Dim Light Melatonin Onset (DLMO) based on the hockey-stick method [47]. For further details on the analytical method and performance characteristic, see Appendix A.

Note: shading in the table represents the distinction between variables. Abbreviations: CES-D, Center for Epidemiological Studies—depression scale; CWS, Cancer Worry Scale; FCS, Fatigue Catastrophizing Scale; MFI, Multidimensional Fatigue Inventory; nmol/L, nanomoles per liter; PSQI, Pittsburgh Sleep Quality Index; RAND-36, medical outcome studies short form; SES-28, self-efficacy scale 28; STAI-6, State-Trait Anxiety Inventory (6); VAS, visual analogue scale; WSAS, Work and Social Adjustment Scale.

**Table 2 cancers-13-04948-t002:** Baseline sociodemographic, clinical, and fatigue characteristics (*N* = 166) ^a^.

Characteristic		No. (%) ^b^			*p*	*N*
		All Survivors	BWL (*n* = 83)	DWL (*n* = 83)		
Age, years						166
	Mean	45.7	46.7	44.8	0.30	
	SD	12.2	11.9	12.5		
Female		99 (59.6)	50 (60.2)	49 (59.0)	0.87	166
Education					0.24	165
	None/primary	2 (1.2)	0 (0.0)	2 (2.4)		
	High school and vocational	85 (51.5)	43 (51.8)	42 (51.2)		
	College or university	78 (47.3)	40 (48.2)	38 (46.3)		
Married or in relationship		130 (78.8)	71 (85.5)	59 (72.0)	0.03	165
Part- or full-time job		85 (51.5)	42 (50.6)	43 (52.4)	0.81	165
Chronotype					0.44	165
	Morning type	29 (35.4)	56 (33.9)	27 (32.5)		
	Evening type	33 (40.2)	74 (44.8)	41 (49.4)		
	No specific type	20 (24.4)	35 (21.2)	15 (18.1)		
Recruitment					0.86	166
	Asked by physician	50 (30.1)	24 (28.9)	26 (31.3)		
	Survey study	98 (59.0)	49 (59.0)	49 (59.0)		
	Applied for participation	18 (10.8)	10 (12.0)	8 (9.6)		
Diagnosis					0.68	166
	HL	138 (83.1)	70 (84.3)	68 (81.9)		
	DLBCL	28 (16.9)	13 (15.7)	15 (18.1)		
Ann Arbor stage					0.64	155
	I	21 (12.7)	10 (12.0)	11 (13.3)		
	II	87 (52.4)	40 (48.2)	47 (56.6)		
	III	25 (15.1)	14 (16.9)	11 (13.3)		
	IV	22 (13.3)	13 (15.7)	9 (10.8)		
Time since diagnosis, years ^c^					0.88	166
	Mean	12.9	13.0	12.9		
	SD	9.9	9.6	10.3		
	2–5 years	41 (24.7)	20 (24.1)	21 (25.3)	0.97	
	5–10 years	50 (30.1)	24 (28.9)	26 (31.3)		
	10–20 years	39 (23.5)	20 (24.1)	19 (22.9)		
	>20 years	36 (21.7)	19 (22.9)	17 (20.5)		
Treatments received						
	Radiotherapy	116 (72.0)	56 (69.1)	60 (75.0)	0.41	161
	Chemotherapy	151 (93.2)	76 (92.7)	75 (93.8)	0.79	162
	Stem cell transplantation	19 (11.8)	8 (9.9)	11 (13.8)	0.45	161
	Total body irradiation ^d^	2 (1.2)	0 (0.0)	2 (2.5)	0.24	162
	Surgery (splenectomy) ^d^	6 (3.7)	3 (3.7)	3 (3.8)	1.0	162
Relapse		25 (15.4)	13 (15.9)	12 (15.0)	0.88	162
Second malignancies		25 (15.7)	13 (15.7)	12 (15.4)	0.91	159
Hyperthyroidism ^d,e^		1 (0.6)	0 (0.0)	1 (1.3)	0.49	156
Hypothyroidism ^e^		36 (23.1)	21 (26.3)	15 (19.7)	0.34	156
Heart complaints, NYHA class 1 or 2		33 (20.8)	19 (23.5)	14 (17.9)	0.39	159
**Fatigue (baseline)**						
VAS					0.09	164
	Mean	6.1	5.9	6.3		
	SD	1.6	1.8	1.4		
MFI general fatigue					0.76	165
	Mean	15.7	15.6	15.8		
	SD	2.7	2.9	2.5		
Work and social restrictions caused by fatigue (WSAS)					0.73	165
	Mean	20.5	20.7	20.2		
	SD	8.2	7.8	8.5		
Sleep medication use		25 (15.2)	11 (13.3)	14 (17.1)	0.49	165

Note: shading in the table represents the distinction between variables. Abbreviations: BWL, bright white light; DWL, dim white light; SD, standard deviation; HL, Hodgkin lymphoma; DLBCL, diffuse large B-cell lymphoma; VAS, visual analogue scale. ^a^ Medical information was available by less than the total number of participants due to missing data in the medical information form completed by treating physician or researcher, ^b^ unless otherwise specified. ^c^ Based on Mann–Whitney Test. ^d^ Based on Fisher’s Exact Test. ^e^ Survivors were included when their medication use was stable for ≥6 months and fatigue complaints remained.

**Table 3 cancers-13-04948-t003:** Light therapy characteristics *.

Characteristic		No. (%) ^a^			*p*	*N*
		All Survivors	BWL (*n* = 83)	DWL (*n* = 83)		
Season LT start					0.94	164
	Autumn	42 (25.6)	23 (27.7)	19 (23.5)		
	Winter	47 (28.7)	23 (27.7)	24 (29.6)		
	Spring	47 (28.7)	23 (27.7)	24 (29.6)		
	Summer	28 (17.1)	14 (16.9)	14 (17.3)		
LT device						
	Litebook Edge	37 (22.6)	18 (21.7)	19 (23.5)		164
	Luminette	127 (77.4)	65 (78.3)	62 (76.5)		164
Days of LT use based on LT diary ^b^					0.52	155
	Mean	22.7	22.5	22.9		
	SD	4.4	4.6	4.0		
	>25 days ^c^	3 (1.9)	0 (0.0)	3 (3.9)	0.13	155
	25 days	58 (37.4)	33 (41.8)	25 (32.9)		
	14–24 days	87 (56.1)	41 (51.9)	46 (60.5)		
	1–13 days (premature stop)	7 (4.5)	5 (6.3)	2 (2.6)		
Time difference sleep end and LT start (min) ^d^					0.13	155
	Mean	25.0	27.4	22.6		
	SD	19.5	22.6	15.3		
Time difference DLMO and LT start (h)					0.17	45
	Mean	11.4	11.1	11.7		
	SD	1.5	1.0	1.9		
	*n*	45	23	22		
Self-reported side effects						
	Headache/nausea	35 (21.6)	21 (25.6)	14 (17.5)	0.21	162
	Feeling agitated ^b^	5 (3.1)	1 (1.2)	4 (5.0)	0.21	162
	Tired eyes	30 (18.5)	11 (13.4)	19 (23.8)	0.09	162
	Change in vision ^b^	8 (4.9)	5 (6.1)	3 (3.8)	0.72	162
	Other self-reported side effects ^e^	15 (9.3)	6 (7.3)	9 (11.3)	0.39	162
Premature stop of LT		13 (7.8)	7 (8.4)	6 (7.2)	0.77	166
Reasons for premature stop ^b^					0.21	13
	Self-reported side effects	7 (53.8)	5 (71.4)	2 (33.3)		
	No time or personal circumstances	6 (46.2)	2 (28.6)	4 (66.7)		

Note: shading in the table represents the distinction between variables. * Two participants never started with light therapy and therefore characteristics of light therapy are reported for 164 participants. Light therapy diaries were completed by 155 participants. DLMO was determined for 45 participants. In total, 162 participants completed the screening on side effects during a telephone call. Abbreviations: BWL, bright white light; DLMO, dim light melatonin onset; DWL, dim white light; LT, light therapy. ^a^ Unless otherwise specified. ^b^ Categorical test results is based on Fisher’s Exact Test. ^c^ Some individuals misinterpreted the study protocol and used light therapy for 28, 30, or 33 days. ^d^ Based on Mann–Whitney Test. ^e^ Other self-reported side effects included worse sleep quality (*n* = 7), feeling more fatigued (*n* = 2), feeling rushed (*n* = 1), shingles (*n* = 1), feeling confused (*n* = 1), sensitive gingiva (*n* = 1), and a dry mouth (*n* = 1).

**Table 4 cancers-13-04948-t004:** Number (percentage) of participants with clinically meaningful improvement based on fatigue assessments.

Outcome		T0-T1 ^a^				T0-T2 ^a^				T0-T3 ^a^			
		No. (%)				No. (%)				No. (%)			
		BWL	DWL	*p* ^b^	OR ^c^	BWL	DWL	*p* ^b^	OR ^c^	BWL	DWL	*p* ^b^	OR ^c^
VAS fatigue													
	Improved	34 (42)	41 (55)	0.11	0.60	25 (35)	37 (52)	0.04	0.50	24 (33)	27 (39)	0.48	0.78
	Not improved	47 (58)	34 (45)			46 (65)	34 (48)			49 (67)	43 (61)		
	*n*	81	75			71	71			73	70		
MFI general fatigue													
	Improved	49 (60)	47 (63)	0.71	0.89	35 (49)	37 (52)	0.68	0.87	36 (49)	40 (57)	0.31	0.71
	Not improved	33 (40)	28 (37)			37 (51)	34 (48)			38 (51)	30 (43)		
	*n*	82	75			72	71			74	70		
WSAS													
	Improved	33 (40)	26 (35)	0.47	1.27	31 (43)	27 (39)	0.59	1.20	31 (42)	29 (41)	0.96	1.02
	Not improved	49 (60)	49 (65)			41 (57)	43 (61)			43 (58)	41 (59)		
	*n*	82	75			72	70			74	70		

Note: shading in the table represents the distinction between variables. Abbreviations: MFI, Multidimensional Fatigue Inventory; OR, odds ratio; VAS, visual analogue scale; WSAS, Work and Social Adjustment Scale. ^a^ T0, baseline; T1, post-intervention; T2, 3 months after light therapy; T3, 9 months after light therapy. ^b^
*p* value of the Pearson chi-square test. ^c^ Odds ratios of 1.5 were considered small, 2.0 as moderate, and 3.0 as large.

## Data Availability

The data underlying this article will be shared on reasonable request to the corresponding author.

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
