# Peer review of "Light Therapy for Cancer-Related Fatigue in (Non-)Hodgkin Lymphoma Survivors: Results of a Randomized Controlled Trial"

_cancers, 2021, doi:10.3390/cancers13194948_

Round 1

Reviewer 1 Report

The authors are to be thanked for the considerable effort expended in conducting such a complex prospective controlled trial in long-term lymphoma survivors.

Overall, the trial design was well done and the primary and secondary outcomes correctly selected. Adherence to the protocol was remarkably high with few missing assessments, a strength that could be highlighted in the discussion section of the manuscript. One explanation could be that the survivors were heavily selected from those with severe chronic fatigue.Supplemental material is very dense and I wonder whose reader will read it extensively!However, few changes can be done that will improve the manuscript. They are listed below according to their appearance in the manuscript.

  1. Abstract, results paragraph: “which only slightly diminished during follow-up” could be expressed differently.
  2. Introduction: two recent publications could be added to ref. #1 and 2: Health Qual Life Outcomes 2019;17:115; Cancer 2019;125(13):2291-2299.
  3. In general, tables are difficult to read. Attention should be paid to better separate the various items. In Table 1, it is hard to relate details listed in column 3 to the variable listed in column 1.
  4. Statistical analysis (L1): please define the effect size as made in ref. #30.
  5. Statistical analysis (L5): Fisher exact test should be used for all comparisons where appropriate.
  6. Statistical analysis (L7-8): justify missing values replacement by average score. Give an example to explain how it was calculated for a given item at a given time.
  7. Statistical analysis: can you relate effect size (L21) to clinically relevant improvement (L23-25)?
  8. Table 3: I understand that N in last column indicates the number of survivors whose assessment has been completed (total minus missing data). However, some figures are misplaced, others are superfluous. Please correct.
  9. Section 3.1 Primary outcomes: give p-values where improvement was statistically significant (L3).
  10. Section 3.1 Primary outcomes: while the effect sizes (ES) are shown in the text (L4, 5, 7), the absolute differences are shown on the graphs (y-axis) in Figure 2. Could you add an ES scale on the right y axis?
  11. Same remark for Figure 3.
  12. Discussion (P14, second paragraph, L11-13): you could reinforce your point by pointing out that the survivors were selected from among those who expressed moderate to high chronic fatigue for more than 10 years on average.
  13. Discussion (P14, fourth paragraph, L11-12): I don’t understand the meaning of this sentence.
  14. Appendix 3: concerns missing questionnaires. Numbers in Tables A3.1 and A3.2 do not match. Total of 169 in Table A3.1 and discrepancies for lines OOOM / OOMO and OOMM. Also pattern MMMM is missing in Table A3.1.

Reviewer 2 Report

Major concerns:

  1. One of the primary outcomes of fatigue was measured by a single item VAS and assessed fatigue at that moment. Although a measure like this may be more efficient in a clinic setting, this does seem like an out of place measure, especially as a primary outcome, in a long-term clinical trial. Given that cancer-related fatigue is a multidimensional outcome that can present in a number of different ways both between and within persons over time, it is unlikely that this nuance would be captured by a singular item assessing fatigue at one particular moment in time. Items of this nature are often heavily influenced by context and other momentary and daily factors, and likely shows variability from moment to moment. For example, research examining fatigue using ambulatory assessment and ecological momentary assessment using similar items and rating scales, indicates that fatigue varies considerably within-persons over the day and over longer periods of time (Banthia, et al., 2006; Curran, Beacham, Andrykowski, 2004). It was promising that the authors included a multidimensional scale of fatigue as well to supplement their primary outcome of fatigue. However, as presented in this manuscript, the listing of this outcome as the first primary outcome is somewhat misleading and likely isn’t capturing fatigue as intended or described. The authors might consider discussing some of the limitations of this measurement approach in the methods section and discussion, and also place more attention on the more comprehensive MFI in their results.
  2. It would be helpful if the authors could provide some justification for not including any inter-treatment measurement of the primary outcomes, especially given that they are interested in examining the mechanisms of light therapy (i.e., sleep and circadian rhythms, cortisol, cytokine expression), most of which display variability from day-to-day, or week-to-week. Perhaps an acknowledgement in the discussion section that a more granular approach to assessing the intervention’s impact during the intervention period would help elucidate the mechanisms underlying this type of fatigue. This may help the reader understand some of the main issues associated with this assessment schedule. Although long-term assessments of this intervention are important, it is also important to understand how and why the intervention is working so it can be modified to be more efficacious, leading to studies that can study the longer term impact. I think that the results of this study indicate that exact dilemma, as even the active component of the intervention (e.g.,, brightness/intensity or wavelength; behavior change) is still not clear, nor is the mechanism by which it can improve fatigue after even a short intervention period of 28-days.
  3. The authors noted that they changed light therapy devices part way through the study because they did not believe the devices were sufficient in their intensity/brightness (lux), though each of the previous trials used the same devices and those studies showed (clinically) meaningful changes on several outcomes. This rationale seems odd, especially part way through a study that is listed as double-blind and leads one to wonder whether the researchers were monitoring the data to check on the intervention effects. Perhaps the sensitivity analyses comparing the types if light therapy devices help to dissuade some of these concerns, but this reads as an odd decision part way through a clinical trial, especially with an established treatment option. Some additional clarification and transparency on this decision is suggested.
  4. Although there is some justification for the use of the dim white light as a comparison group outlined in the protocol paper (used as a comparator in Alzheimer’s disease, though there is no published work on this), it is still unclear why the authors chose this as their comparator instead of a dim red light as was used in the other previous studies of light therapy for cancer-related fatigue. Are the authors then suggesting that brightness (lux) alone is the active component of light therapy, and wavelength has no impact? Although this is a reasonable suggestion, the results (and the authors’ conclusion in the discussion) seem to indicate that wavelength is indeed important as there were very few differences on the primary and secondary outcomes between the two light therapy conditions. Although the authors note that other studies did not seem to find superiority of BWL to DRL on some dimensions of fatigue, there were other notable limitations to these studies (i.e., smaller sample sizes) that likely preclude their ability to detect meaningful differences in the conditions on some subscales due to a lack of power. It is suggested that the authors include a more robust justification for their selection of their comparator in the methods section, as it is also not sufficiently addressed in the protocol paper.

Minor concerns:

  1. The authors might consider discussing some of the limitations of collecting cortisol on only one day at pre- and post-intervention (see Segerstrom, Boggero, Smith, & Sephton, 2014), and how this may have impacted their results.
  2. Most of the reference numbers in Table 1 are not correctly attributed to the articles in the reference list, so it is suggested that all references throughout the paper are reviewed and corrected as necessary.
  3. Table 1, as formatted in the reviewer proof, is very difficult to read with no horizontal lines between scale types. Similarly, the graphs as presented are not interpretable with black and white or greyscale as the lines look like the same color.

Round 2

Reviewer 2 Report

The authors have done a nice job of addressing the reviewer comments appropriately and the changes to the manuscript have improved the paper. I have no other comments or concerns.